# Unraveling Hematotoxicity of α-Amanitin in Cultured Hematopoietic Cells

**DOI:** 10.3390/toxins16010061

**Published:** 2024-01-22

**Authors:** Willemien F. J. Hof, Miranda Visser, Joyce J. de Jong, Marian N. Rajasekar, Jan Jacob Schuringa, Inge A. M. de Graaf, Daan J. Touw, Bart G. J. Dekkers

**Affiliations:** 1Department of Clinical Pharmacy and Pharmacology, University Medical Center Groningen (UMCG), 9713 GZ Groningen, The Netherlands; w.f.j.hof@umcg.nl (W.F.J.H.);; 2Department of Experimental Hematology, University Medical Center Groningen, University of Groningen, 9713 GZ Groningen, The Netherlands

**Keywords:** *Amanita phalloides*, α-amanitin, hematopoietic cell lines, CD34+ stem cells, apoptosis

## Abstract

*Amanita phalloides* poisonings account for the majority of fatal mushroom poisonings. Recently, we identified hematotoxicity as a relevant aspect of *Amanita* poisonings. In this study, we investigated the effects of the main toxins of *Amanita phalloides*, α- and β-amanitin, on hematopoietic cell viability in vitro. Hematopoietic cell lines were exposed to α-amanitin or β-amanitin for up to 72 h with or without the pan-caspase inhibitor Z-VAD(OH)-FMK, antidotes N-acetylcysteine, silibinin, and benzylpenicillin, and organic anion-transporting polypeptide 1B3 (OATP1B3) inhibitors rifampicin and cyclosporin. Cell viability was established by trypan blue exclusion, annexin V staining, and a MTS assay. Caspase-3/7 activity was determined with Caspase-Glo assay, and cleaved caspase-3 was quantified by Western analysis. Cell number and colony-forming units were quantified after exposure to α-amanitin in primary CD34+ hematopoietic stem cells. In all cell lines, α-amanitin concentration-dependently decreased viability and mitochondrial activity. β-Amanitin was less toxic, but still significantly reduced viability. α-Amanitin increased caspase-3/7 activity by 2.8-fold and cleaved caspase-3 by 2.3-fold. Z-VAD(OH)-FMK significantly reduced α-amanitin-induced toxicity. In CD34+ stem cells, α-amanitin decreased the number of colonies and cells. The antidotes and OATP1B3 inhibitors did not reverse α-amanitin-induced toxicity. In conclusion, α-amanitin induces apoptosis in hematopoietic cells via a caspase-dependent mechanism.

## 1. Introduction

*Amanita phalloides*, also known as death cap, accounts for the majority of fatal mushroom poisonings and is considered one of the most poisonous mushrooms worldwide [1,2,3]. Growing interest in collecting wild mushrooms over the past few decades has led to an increase in mushroom poisonings [2,3]. *A. phalloides* contains three groups of cyclic peptide toxins: amatoxins, phallotoxins, and virotoxins. Of these, amatoxins are considered to be mainly responsible for the toxic effects of the mushroom. The main amatoxins are α-amanitin and β-amanitin [4]. Remarkably, amatoxins are very stable in aqueous solutions and resistant to cooking, freezing and drying, and acid or enzyme degradation. Consequently, amatoxins are not destroyed during food preparation or after ingestion [5].

Amatoxins have a high bioavailability and are rapidly absorbed from the gastrointestinal tract. Initial symptoms include gastrointestinal complaints [6]. After absorption, amatoxins cause liver and kidney toxicity [7,8]. After a latent period, poisoning may result in hepatorenal failure, jaundice, seizures, and ultimately death in more severe cases. Acute liver failure is a particular characteristic of amatoxin poisoning [6]. Uptake of amatoxins into hepatocytes has been attributed to the organic anion-transporting polypeptide 1B3 (OATP1B3), which is present in the sinusoidal membrane of the hepatocytes [9]. The exact mechanism of its toxicity remains elusive. Binding of RNA polymerase II (RNA PII) by α-amanitin is considered to be the main toxicodynamic event. The formation of this complex leads to deficient protein synthesis and eventually cell death [10]. However, other mechanisms have also been proposed, involving reactive oxygen species (ROS), tumor necrosis factor-α (TNF-α), and various pathways of apoptosis [11,12,13,14].

Therapy consists of antidote treatment, maintenance of vital functions, and supportive measures. An emergency liver transplantation may be indicated in severe cases. However, a mortality rate of 10–40% is still reported for adult amatoxin poisonings in Europe [15,16,17]. Frequently used antidotes include N-acetylcysteine, silibinin, and benzylpenicillin [17]. The mechanisms of these antidotes are not completely elucidated but appear to involve inhibition of oxidative stress and/or the blockade of OATP1B3 [18]. Other known OATP1B3 inhibitors are the immunosuppressant cyclosporin and the antibiotic rifampicin [9], which have been shown to inhibit amanitin-induced cellular damage [9,19]. A recent study evaluated the effects of N-acetylcysteine, silibinin, and benzylpenicillin on clinical outcomes of amatoxin poisonings. An overall survival rate of 84% was found among all patients receiving one antidote, a combination of two or three of these antidotes, other treatment, or no treatment. In comparison, 59% of patients survived when treated with supportive care alone. This study also found that monotherapy with silibinin or benzylpenicillin was associated with a higher survival rate than supportive care only [17]. In line with this, N-acetylcysteine, silibinin, and benzylpenicillin as monotherapy or combined therapies have been shown to be options in the treatment of amatoxin poisonings [20,21,22,23].

Recently, we demonstrated that in addition to liver and renal toxicity, human *A. phalloides* poisonings are associated with hematotoxicity [24]. In these patients, we found that during admission, *A. phalloides* poisoning is associated with decreases in hemoglobin, hematocrit, leukocytes and platelets [24]. In the current study, we investigate these findings in vitro and mechanistically for the first time. The aim of the current study is to investigate the effects of α-amanitin on the cell viability of multiple hematopoietic cell lines and primary CD34+ stem cells in vitro. The toxic effects of β-amanitin are also established. The mitigating effects of commonly used antidotes, N-acetylcysteine, silibinin, and benzylpenicillin, and specific inhibitors of the OATP1B3 transporter, rifampicin and cyclosporin, are evaluated. Finally, the role of caspase-dependent apoptosis in the decrease in cell viability is studied.

## 2. Results

### 2.1. α-Amanitin and β-Amanitin Decrease Viability of HL60 Cells

The effects of α-amanitin on cell viability were investigated by trypan blue exclusion and a 3-(4,5-dimethylthiazol-2-yl)-5(3-carboxymethoxyphenyl)-2-(4-sulfopheny)-2H-tetrazolim (MTS) assay. In the trypan blue exclusion assay, α-amanitin concentration-dependently decreased the number of viable cells after 72 h of exposure (Figure 1A). This effect was significant for 1 μM (*p* < 0.05), 3 μM, and 10 μM α-amanitin (*p* < 0.001). The average percentage of viable cells remaining compared to the control was 87 ± 11% for 1 μM α-amanitin, 78 ± 16% for 3 μM α-amanitin, and 12 ± 8% for 10 μM α-amanitin. Preliminary experiments showed a maximum effect of α-amanitin on cell viability after 72 h of exposure. No significant effects of exposure to α-amanitin concentrations up to 3 µM were observed after 24 and 48 h. The maximum effect of α-amanitin was comparable to the effect of cyclophosphamide (10 mg/mL), which was used as a positive control. The results of the MTS assay show a similar effect to that observed for the trypan blue exclusion assay (Figure 1B). Both α-amanitin and β-amanitin decreased MTS reduction in a concentration-dependent fashion. However, the effect of α-amanitin was more significant than that of β-amanitin, especially at the highest concentrations. At 10 μM α-amanitin, only 12 ± 13% MTS reduction remained, while exposure to β-amanitin at this concentration resulted in 58 ± 38% MTS reduction (*p* < 0.001).

### 2.2. α-Amanitin Decreases Viability of Multiple Hematopoietic Cell Lines

To investigate whether the observed effects were also of relevance for other hematopoietic cells, the influence of α-amanitin on the cell viability of five additional hematopoietic progenitor cell lines was subsequently determined using the MTS assay. As shown in Figure 2, α-amanitin caused a concentration-dependent decrease in MTS reduction in all cell lines investigated. Significant effects were observed in HL60 cells for ≥3 μM α-amanitin (*p* < 0.001) and in MV411 cells for ≥0.3 μM α-amanitin (*p* < 0.01, Figure 2A). In THP1, Jurkat, and K562 cells, MTS reduction was significantly decreased at concentrations higher than 1 μM α-amanitin (*p* < 0.001, Figure 2B). In SUDHL6 cells, a significant decrease in MTS reduction was observed for ≥3 μM (*p* < 0.01). The IC_50_ values of α-amanitin in the cell lines were 0.59 ± 0.07 μM for the MV411 cell line, 0.72 ± 0.09 μM for the THP1 cell line, 0.75 ± 0.08 μM for the Jurkat cell line, 2.0 ± 0.18 μM for the K562 cell line, 3.6 ± 1.02 μM for the SUDHL6 cell line, and 4.5 ± 0.73 μM for the HL60 cell line. HL60 cells and SUDHL6 cells were significantly less sensitive to α-amanitin than the other cell lines (Appendix A).

### 2.3. Hit-and-Run Effect of α-Amanitin

To mimic a hypothesized hit-and-run effect of α-amanitin (due to its short half-life in vivo after oral absorption), HL60 cells were exposed to α-amanitin (0–10 µM) for 4 h or 16 h. After this, cells were washed, and culture medium was replaced by fresh medium without α-amanitin for the remainder of the 72 h. Schematic representations of the experimental protocols are shown in Appendix A. After 72 h, viability was determined using the MTS assay. α-Amanitin (10 μM) decreased MTS reduction significantly after only 4 h of exposure compared to the control (0 μM, *p* < 0.01; Appendix A). MTS reduction was significantly more decreased after 72 h of continuous exposure to α-amanitin than after 4 h of exposure to 3 μM and 10 μM α-amanitin followed by culture in fresh medium (*p* < 0.001). After 16 h of exposure to α-amanitin, both 3 μM and 10 μM α-amanitin significantly decreased MTS reduction (*p* < 0.001). For 10 μM α-amanitin, no significant difference was observed between 16 h of exposure followed by culture in fresh medium and 72 h of continuous exposure.

### 2.4. α-Amanitin Decreases Viability of Primary CD34+ Stem Cells

To determine whether α-amanitin also decreased viability in primary hematopoietic cells, a colony-forming cell (CFC) assay using CD34+ stem cells was performed. In this assay, α-amanitin was found to decrease the number of CD34+ stem cell colonies in a concentration-dependent manner as well (Figure 3A). At concentrations of 1, 3 and 10 μM, α-amanitin significantly decreased the total number of colonies compared to the control (0 μM, *p* < 0.001). The corresponding IC_50_ value was 1.0 ± 0.28 μM, similar to the IC_50_ in THP1 and Jurkat cells, but significantly lower than the IC_50_ value in SUDHL6 and HL60 cells (*p* < 0.001). No significant changes in the ratio of different types of colonies were found, suggesting that the α-amanitin-induced decrease in colony-forming units was not due to toxicity towards a specific subset. In line, α-amanitin decreased the total number of cells, also concentration-dependently (Figure 3B). Cell number was significantly reduced after exposure to 1 μM (*p* < 0.01), 3 μM, and 10 μM α-amanitin (*p* < 0.001). The calculated IC_50_ value was 0.71 ± 0.21 μM α-amanitin, in line with the IC_50_ value for the colony-forming units.

### 2.5. Antidotes Do Not Prevent α-Amanitin-Induced Hematotoxicity

To study whether antidotes prevented the reduction in cell viability in response to α-amanitin, the cells were pretreated with the commonly used antidotes N-acetylcysteine, benzylpenicillin, and silibinin. The antidotes did not significantly prevent α-amanitin-induced toxicity (Figure 4). Combinations of the compounds showed similar effects to those of the individual compounds. No effect of the solvent (DMSO) was observed. Interestingly, silibinin augmented the toxic effect of 3 μM α-amanitin significantly (*p* < 0.001). Additional experiments confirmed that silibinin augmented α-amanitin-induced toxicity (Appendix A). The calculated IC_50_ value for α-amanitin with silibinin (2.03 ± 0.04 μM) was significantly lower than the IC_50_ value for α-amanitin without silibinin (4.18 ± 0.68 μM). N-acetylcysteine and benzylpenicillin did not significantly influence α-amanitin-induced toxicity. The OATP1B3 inhibitors rifampicin and cyclosporin did not significantly affect α-amanitin-induced toxicity either.

### 2.6. Increased Caspase Activity Is Involved in α-Amanitin-Induced Cell Death

To study the mechanism involved in the α-amanitin-induced reductions in cell viability, HL60 cells were stained for annexin V and propidium iodide (PI). In addition to decreasing cell viability, α-amanitin increased the percentage of early apoptotic, late apoptotic, and necrotic cells (Figure 5A). These effects were both concentration- and time-dependent. The number of early apoptotic cells was significantly increased after 48 h by 10 μM α-amanitin (*p* < 0.01, Appendix A).

As increase in caspase activity is the initial process prior to cell death. A change in caspase-3/7 activity was expected to occur before the increase in apoptotic cells. Therefore, caspase-3/7activity was measured after 24 h. At this moment, no cell death had been observed yet. Significant increases in caspase-3/7 activity were observed as soon as 24 h of exposure to α-amanitin (3 and 10 µM, Figure 5B). These conditions showed 157 ± 5% and 282 ± 6% activity relative to the control, respectively (*p* < 0.001). Moreover, α-amanitin increased cleaved caspase 3 levels in 24 h. This increase reached statistical significance at concentrations of 1 μM (*p* < 0.01) and 10 μM (*p* < 0.001) α-amanitin compared to the control (Figure 5C). The pan-caspase inhibitor Z-VAD(OH)-FMK significantly prevented a decrease in MTS reduction in response to 10 µM α-amanitin (*p* < 0.01, Figure 5D). A similar, but not statistically significant, effect was observed for 3 µM α-amanitin.

## 3. Discussion

In this study, we demonstrated for the first time that α-amanitin had a concentration-dependent and time-dependent toxic effect on multiple human hematopoietic progenitor cells lines and primary human CD34+ stem cells. Short-term exposure appeared sufficient to reduce viability at 72 h. α-Amanitin reduced cell viability both via apoptosis and necrosis. No protective effects on cell viability or the number of cells were observed in our experimental setup for the commonly used antidotes N-acetylcysteine, silibinin, and benzylpenicillin. Pharmacological inhibition of caspase activity partially prevented α-amanitin-induced toxicity.

Apoptosis has been shown to play an important role in α-amanitin-induced liver injury in dog primary hepatocytes and human primary hepatocytes [13,25]. Necrosis was also established in dog hepatocytes and in vivo in the liver of pigs [26,27]. Furthermore, increased cleaved caspase-3 and a concentration-dependent and time-dependent effect were demonstrated in mouse liver Hepa1-6 cells [28]. In line with these studies, α-amanitin-induced toxicity in our hematopoietic cell lines was found to be dependent on apoptosis and necrosis. Cell lines from multiple differentiation pathways were used, illustrating that α-amanitin-induced toxicity affects an array of hematopoietic cell types. The primary human CD34+ stem cells we used provide an even more realistic model for α-amanitin-induced hematotoxicity. These results underline that primary cells are more sensitive to α-amanitin than HL60 cells.

Multiple mechanisms of α-amanitin-induced cell death are described in the literature, specifically regarding apoptosis. p53- and caspase-3-dependent apoptosis was detected in human hepatocytes exposed to 2 μM α-amanitin for 24 h [13]. RNAPII inhibition by α-amanitin was shown to elicit translocation of cytoplasmic p53 to mitochondria [29]. This caused permeabilization of the mitochondrial membrane, release of cytochrome C into the cytosol, and initiation of the intrinsic apoptotic pathway [29]. Other mechanisms of apoptosis may also be involved, including ROS formation and TNF-α upregulation [27,28]. Our results show that the inhibition of apoptosis prevents a significant amount of α-amanitin-induced toxicity, especially at a high α-amanitin concentration.

N-acetylcysteine, silibinin, and benzylpenicillin are commonly used antidotes in the treatment of *A. phalloides* poisonings [30]. The exact mechanism of action of these antidotes is currently unknown, but the scavenging of free radicals, increased glutathione synthesis, and OATP1B3 inhibition are considered possible targets for these antidotes [9,18,30,31,32,33]. In our study, however, no effects of these antidotes were observed on α-amanitin-induced hematotoxicity. The OATP1B3 inhibitors cyclosporin and rifampicin [9] also had no significant effect on toxicity of α-amanitin. OATP1B3 is not or to a very limited extent expressed in the cell lines used in our study [34]. Therefore, it is unlikely that α-amanitin is taken up via this transporter and that the antidotes would have an effect via the inhibition of this transporter. Moreover, the absence of OATP1B3 did not completely prevent α-amanitin-induced toxicity in the original study of Letschert et al. [9], suggesting alternative uptake mechanisms of α-amanitin in addition to the established route via OATP1B3. In short, the mechanisms of uptake and toxicity of α-amanitin in hematopoietic cells remain to be elucidated and require further research. Suppression of growth of the HL60 cell line by silibinin is in line with previous studies [35], whereas no effects have been observed for N-acetylcysteine and benzylpenicillin [36,37]. Based on these studies, a synergistic effect of silibinin on α-amanitin-induced toxicity could be hypothesized, but the exact underlying mechanism is unknown.

Amatoxins are detected in urine within 90–120 min after ingestion according to a previous study [7]. Only trace amounts are usually found in plasma 24 h after ingestion, and within 48 h, amatoxins are completely eliminated from plasma in most patients, despite causing serious symptoms [38,39,40]. This suggests a hit-and-run effect of α-amanitin. Our results confirm this, showing that a high concentration of α-amanitin exerts its full toxic effect within the first 16 h. This poses a problem for patients, as the first symptoms of poisoning usually only appear after a latency period of 8 to 24 h [41]. Consequently, a large part of the damage may already be done before treatment can start.

The range of α-amanitin concentrations used in our experiments were based on an estimation of clinically relevant concentrations in human poisonings. Clinically relevant concentrations of α-amanitin for the experiments were estimated using toxicokinetic data found in beagle dogs exposed to *A. exitialis* [42,43]. This mushroom contains similar toxins to *A. phalloides* according to previous research [42]. An allometric scaling factor of 0.75 for volume of distribution and clearance from dogs to humans was used [44]. Accounting for these factors and the increased body weight, doses of α-amanitin equivalent to those administered to the beagle dogs were calculated for humans. These doses, approximately 0.3–1.1 mg of α-amanitin, would result in peak plasma concentrations of roughly 0.06–0.13 μM in humans. In humans, the LD_50_ of α-amanitin is estimated in the literature to be 0.1 mg/kg body weight [45]. *A. phalloides* mushrooms have been shown to contain 13–17% α-amanitin in other studies [46]. This means only a single large mushroom of 40 g or a few smaller ones would expose an adult to a potentially lethal dose α-amanitin [46]. In a previous study, the peak plasma concentration in dogs was found to increase proportionally to the quantity α-amanitin ingested [42]. Therefore, an adult (70 kg) ingesting 7 mg of α-amanitin is estimated to obtain a peak plasma concentration of about 0.6–1.7 μM α-amanitin. However, case reports have established that patients may consume larger quantities of mushrooms [47,48]. Moreover, patients with a lower body weight, including children, ingest a relatively larger dose, which may result in a higher plasma concentration of α-amanitin [20,41]. Case reports also indicate plasma concentrations up to 0.13–0.21 µM after 24–48 h [39], suggesting much higher peak concentrations. Based on these considerations, concentrations between 0.1 and 10 μM, as used in the present study, may be clinically relevant. The IC_50_ values of α-amanitin found in primary human hematopoietic cells and colonies in this study were approximately 0.7 µM and 1.0 µM, respectively. These concentrations fall well within the range of estimated relevant plasma concentrations. Another study found an approximate 50% decrease in MTT absorption after exposing primary human hepatocytes to 2 µM α-amanitin for 24 h [13]. This may point towards similar sensitivity to α-amanitin toxicity between human hematopoietic cells and hepatocytes.

β-Amanitin was significantly less potent than α-amanitin. In line with this, LD_50_ values of 10 μg/mL for β-amanitin and 1 μg/mL for α-amanitin have been reported previously in MCF-7 cells after 36 h of incubation [49]. However, one study found significantly higher plasma concentrations of β-amanitin than α-amanitin in four out of nine patients [50]. Therefore, β-amanitin may still contribute to the total toxicity in a clinically relevant way. We also found that β-amanitin toxicity may depend on the physiological environment. The toxicity of α-amanitin and β-amanitin was investigated using two batches of HL60 cells. Both batches exhibited similar proliferation rates, but we observed that the medium of one batch HL60 cells changed color more rapidly over time. This is a sign of acidity and may be caused by a higher metabolic rate of the cells. This may occur when cancer cells in a nutrient-rich environment metabolize more nutrients than they require for proliferation [51]. Remarkably, the toxicity of the two highest concentrations of β-amanitin was increased in the more acidic environment. By contrast, no change was observed in the cell viability of the controls or the toxicity of α-amanitin and lower concentrations of β-amanitin. These results suggest that α-amanitin is equally potent in both a neutral and acidic environment, while β-amanitin may be more potent in an acidic environment. We hypothesize that the toxicity of β-amanitin is diminished at a neutral pH by deprotonation of the carboxylic acid group (pKa 5) of β-amanitin. α-Amanitin has a primary amide group at this position (pKa 18). The α-amanitin group maintains a neutral charge at both neutral pH and a lowered pH. However, the β-amanitin residue is deprotonated at pH 7, and acidification increases the protonated fraction. Previous research found that that the Cα atom of this residue is involved in a hydrophobic interaction with residue His1085 of RNAP II [52]. His1085 is located in the trigger loop of RNAP II. By forming these hydrophobic interactions, α-amanitin inhibits trigger loop movement in RNAP II and thus the incorporation of nucleotides in the RNA [52]. A charged residue may disturb this hydrophobic interaction with RNAP II, limiting the toxic effect of β-amanitin at a neutral pH.

If this hypothesis is correct, alkaline diuresis via sodium bicarbonate administration may be used to treat patients. β-Amanitin is suitable for alkaline diuresis, as amatoxins are excreted renally unchanged, distributed in the extracellular fluid and minimally protein-bound [53]. The mean urine pH (pH 6.25) is also in the range of the pKa of the relevant β-amanitin residue, so alkalinization would significantly increase the deprotonated fraction in urine [54]. Increased excretion of β-amanitin through forced alkaline diuresis may decrease the hematotoxicity of *A. phalloides*, as β-amanitin accounts for a considerable part of the total amanitin content in *A. phalloides*. Concentrations of β-amanitin similar to or significantly larger than the concentration of α-amanitin have been measured [4,46,55]. However, the literature on the toxicokinetics of both α-amanitin and β-amanitin is still very limited. Further research is required to support this hypothesis.

## 4. Conclusions

This in vitro study shows significant toxicity of α-amanitin on hematopoietic cell lines and CD34+ stem cells. In addition, the usual antidotes and specific inhibitors of OATP1B3 do not protect cells against α-amanitin toxicity. Further, β-amanitin demonstrates higher toxicity at a lower pH, although further research is required to show a direct causal relationship between increased toxicity and the altered pH, and not any metabolic changes. If future research confirms the influence of pH on toxicity of β-amanitin, this would open avenues for further investigation towards a potential clinical protective effect of increased excretion by forced alkaline diuresis, for example. Moreover, the observed absorption of α-amanitin in cells without OATP1B3 warrants more research into still-undiscovered uptake mechanisms.

## 5. Materials and Methods

Additional detailed information on the materials and methods is provided in the Appendix A.

### 5.1. Materials

The hematopoietic cells lines that were used were HL60 (promyelocytic leukemia), SUDHL6 (diffuse histiocytic lymphoma), and THP1 (acute monocytic leukemia) cells, MV411 (biphenotypic B-myelomonocytic leukemia), K562 (erythroleukemia) cells, and Jurkat (T-cell leukemia) cells. All cell lines and primary human CD34+ stem cells were a kind gift of the Department of Experimental Hematology, University Medical Center Groningen (UMCG). Dulbecco’s Modified Eagle Medium (DMEM), penicillin–streptomycin (PenStrep), Roswell Park Memorial Institute medium (RPMI), gentamycin, and phosphate-buffered saline were purchased from Gibco (Grand Island, NY, USA). Fetal bovine serum (FBS), α-amanitin, β-amanitin, trypan blue solution (catalog number G3580), N-acetylcysteine, and silibinin were purchased from Sigma-Aldrich (St. Louis, MO, USA). At Promega (Madison, WI, USA), the CellTiter 96^®^ AQueous One Solution Reagent for the MTS assay (catalog number T8154) and the Caspase-Glo 3/7 assay kit were purchased. The pan-caspase inhibitor Z-VAD(OH)-FMK was from Selleck Chemicals (Houston, TX, USA). Benzylpenicillin was purchased at Sandoz (Basel, Switzerland), rifampicin at Alsachim (Illkirch-Graffenstaden, France), and cyclosporin at Cerilliant (Round Rock, TX, USA). At Biolegend (San Diego, CA, USA), the FITC Annexin V Apoptosis Detection Kit with propidium iodide was purchased. Components for RIPA lysis buffer (50 mM Tris-HCl pH 8.0, 150 mM sodium chloride, 1% Igepal Ca630, 0.5% sodium deoxycholate, 1% sodium dodecyl sulphate), SDS precasted polyacrylamide gels, and nitrocellulose membranes were obtained from Bio-Rad (Hercules, CA, USA). Protease inhibitor cocktail was purchased from Thermo Fisher (Waltham, MA, USA). Primary antibody (cleaved caspase 3, rabbit antibody) was from Cell Signaling Technology (Danvers, MA, USA). The secondary antibody (GARP 74 polyclonal goat anti-rabbit immunoglobulins/HRP) was from Dako (Santa Clara, CA, USA). Western Lightning Ultra mixture was purchased from Perkin Elmer (Waltham, MA, USA). All other chemicals were of analytical grade. The NovoCyte Quanteon flow cytometer was from Agilent (Santa Clara, CA, USA). NovoExpress software version 1.6.1 was used for analysis.

### 5.2. Cell Culture

HL60, SUDHL6, and THP1 cells were cultured in DMEM with 20% FBS and 1% PenStrep. MV411 and K562 cells were cultured in DMEM with 10% FBS and 1% PenStrep. Jurkat cells were cultured in RPMI with 10% FBS and 1% PenStrep. All cell lines were cultured at 5% CO_2_, 37 °C and split into fresh medium every 2–3 days, with the exception of THP1 cells being split into fresh medium every 2–3 weeks.

### 5.3. Cell Number and Viability

Effects on cell viability were studied by trypan blue exclusion and MTS conversion assays. Assays were performed according to the manufacturer’s instructions. For the trypan blue exclusion assay, HL60 cells were plated in duplicate or triplicate on 6-well plates (200,000 cells/well) for 72 h in the absence or presence of α-amanitin (0–10 µM). After 72 h, cells were incubated with trypan blue solution (0.4%) and counted using a Bürker-Türk hemocytometer.

Similarly, MTS conversion was determined after 72 h using CellTiter 96^®^ AQ_ueous_ One Solution Reagent. To this aim, HL60, SUDHL6, THP1, Jurkat cells (20,000 cells/well), MV411 cells (10,000 cells/well), and K562 cells (5000 cells/well) were incubated in transparent 96-well plates for 72 h in the absence or presence of α-amanitin or β-amanitin (0–10 µM, both). MV411, K562, and Jurkat cells were plated in quadruplicate, HL60 in quadruplicate or triplicate, SUDHL6 cells in triplicate, and THP1 cells in duplicate. When applied, the pan-caspase inhibitor Z-VAD(OH)-FMK (100 μM), N-acetylcysteine (1 mM), silibinin (30 μM), benzylpenicillin (1 mM), rifampicin (10 μM), and/or cyclosporin (3 μM) were present during the entire incubation period. Conversion of MTS into its reduced form by mitochondrial cytochromes was measured at 490 nm.

### 5.4. Annexin V and PI Analysis

HL60 cells were incubated in triplicate for 24, 48, and 72 h with α-amanitin (0–10 µM) in transparent 96-well plates at a density of 20,000 cells/well. The staining solution was freshly prepared by mixing FITC Annexin V, PI Solution, and Annexin V Binding Buffer in ultrapure water in a ratio of 1:1:32:46. Throughout the entire staining, this solution was protected from light. Plated cells were centrifuged at 350 g for 5 min and washed once with cold phosphate-buffered saline (PBS, pH 7.4). Cells were resuspended in PBS with staining solution at room temperature. Final concentrations of FITC-Annexin V and PI were 0.28 µg/mL and 1.6 µg/mL, respectively. The stained samples were analyzed by flow cytometry. Cells were excited by a 488 nm laser, and detection was performed within 505–560 nm (FITC-Annexin V) and within 595–642 nm (PI). Cells were counted and divided into viable, early-apoptotic, late-apoptotic, and necrotic subpopulations.

### 5.5. Colony-Forming Cell Formation Assays

CFC assays were performed as described earlier [56]. In short, CD34+ stem cells were incubated in 35 mm Petri dishes (500 cells/dish in CFC-mix) in the absence and presence of α-amanitin (0–10 µM) for 14 days, after which stem cell colonies were counted using the colony-forming unit assay. Subsequently, samples were washed with PBS, and total cells were counted using a hemocytometer.

### 5.6. Caspase Activation

The Caspase-Glo 3/7 assay kit was used for detection of caspase activation in HL60 cells according to the manufacturer’s instructions. Cells (20,000 cells/well) were plated in triplicate on white 96-well plates for 24 h in the absence and presence of α-amanitin. Generation of luminescence was measured with a Synergy H4 microplate reader.

Cleaved caspase-3 was quantified by Western blot analysis. HL60 cells were incubated for 24 h on 6-well plates (500,000 cells/mL, 1,000,000 cells/well) in the absence and presence of α-amanitin. Total protein was isolated. Equal amounts were separated on SDS polyacrylamide gels and transferred to nitrocellulose. Bands were visualized using standard immunoblotting techniques, normalized for total protein, and normalized to α-amanitin (10 μM).

### 5.7. Statistical Analysis

Data are presented as mean ± SD. Statistical significance was determined by a one-way ANOVA followed by a Dunnett’s multiple comparisons test or a two-way ANOVA followed by a Tukey’s multiple comparisons test, when appropriate. Dunnett’s multiple comparisons test was used to compare a treatment to its control. Tukey’s multiple comparisons test was used to compare multiple treatments. Data were considered statistically significant when *p* < 0.05.

## Figures and Tables

**Figure 1 toxins-16-00061-f001:**
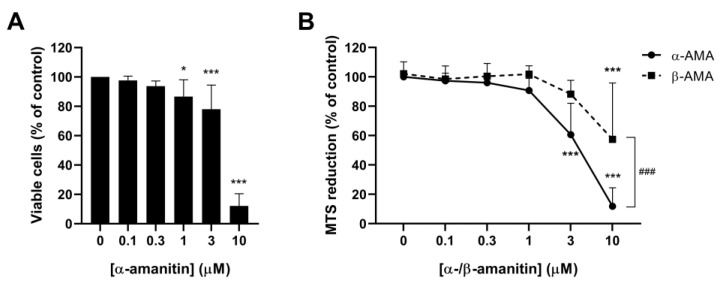
α- and β-amanitin reduce viability of HL60 cells. (**A**) Number of viable cells remaining as assessed by trypan blue exclusion; (**B**) Effects of α-amanitin and β-amanitin on MTS reduction in HL60 cells after 72 h of exposure. Data are presented as a percentage of the control (0 μM α-amanitin). Bars represent mean ± standard deviation (SD), n = 3–7. * *p* < 0.05, *** *p* < 0.001 compared to the control (0 μM), ^###^ *p* < 0.001 compared to equal concentration of β-amanitin.

**Figure 2 toxins-16-00061-f002:**
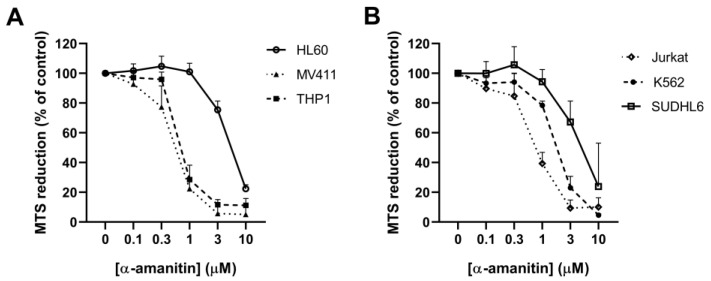
α-Amanitin reduces viability of multiple hematopoietic cell lines. Effects of α-amanitin on MTS reduction in (**A**) HL60, MV411, THP1, (**B**) Jurkat, K562, and SUDHL6 cell lines after incubation for 72 h. Data represent mean ± SD of 3–6 experiments.

**Figure 3 toxins-16-00061-f003:**
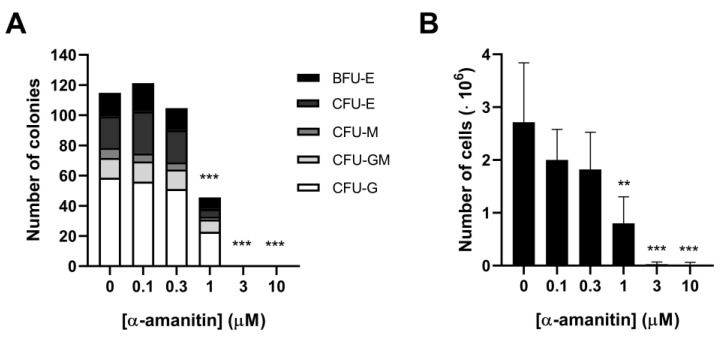
α-Amanitin reduces viability of primary CD34+ stem cells. (**A**) Total number of colonies with a subdivision in the 5 different colonies observed in the CFU assay (n = 3). BFU-E: burst-forming unit-erythroid. CFU-E: colony-forming unit-erythroid. CFU-M: colony-forming unit-macrophage. CFU-GM: colony-forming unit-granulocyte/macrophage. CFU-G: colony-forming unit-granulocyte; (**B**) the cell number (million cells) of CD34+ stem cells treated with increasing concentrations of α-amanitin (n = 2–4). ** *p* < 0.01, *** *p* < 0.001 compared to the control (0 µM).

**Figure 4 toxins-16-00061-f004:**
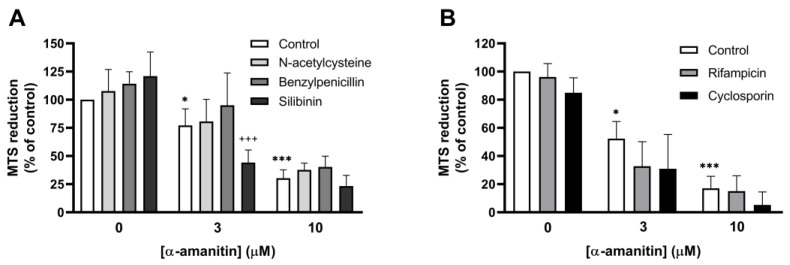
Commonly used antidotes do not prevent α-amanitin induced toxicity. Influence of (**A**) N-acetylcysteine (1 mM), benzylpenicillin (1 mM), silibinin (30 μM), (**B**) rifampicin (10 μM), and cyclosporin (3 μM) on α-amanitin-induced toxicity in HL60 cells. Data represent mean ± SD of 3–9 experiments. * *p* < 0.05, *** *p* < 0.001 compared to the control (0 μM), ^+++^ *p* < 0.001 compared to α-amanitin (3 µM).

**Figure 5 toxins-16-00061-f005:**
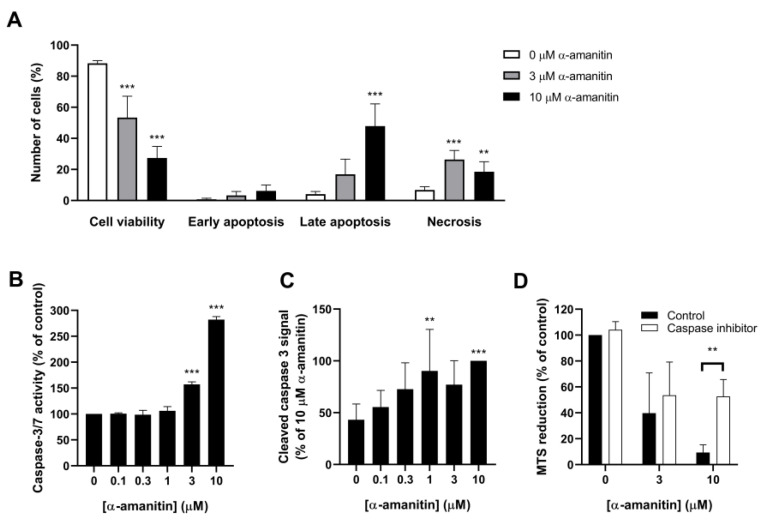
Apoptosis contributes to α-amanitin-induced cell death in HL60 cells. (**A**) Annexin V/PI analysis after 72 h exposure to α-amanitin; (**B**) caspase-3/7 activity as determined by caspase-Glo assay after 24 h of incubation with α-amanitin; (**C**) cleaved caspase 3 after 24 h as measured by Western analysis; (**D**) MTS reduction after 72 h incubation with α-amanitin. Cells were pre-incubated with Z-VAD(OH)-FMK (100 µM). Cells without caspase inhibitor were pre-incubated with the vehicle (DMSO 1%). Data represent mean ± SD of 4–6 experiments. ** *p* < 0.01, *** *p* < 0.001 compared to the control (0 μM).

## Data Availability

Upon reasonable request, and subject to review, the authors will provide the data that support the findings of this study.

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
