# Peer review of "Unraveling Hematotoxicity of α-Amanitin in Cultured Hematopoietic Cells"

_toxins, 2024, doi:10.3390/toxins16010061_

Round 1
Reviewer 1 Report
Comments and Suggestions for Authors
The presented work concerns a new toxic effect of α-amanitin, namely hematotoxicity, which has not been described often so far. The problem is very interesting and the authors implemented a well-planned experimental approach, trying not only to prove the effect on a number of different hematopoetic cell lines, but also to find the potential mechanism of action and finally to verify the use and effectiveness of known hepatoprotectants. In my opinion the work is interesting and written in good English. The experiments are properly described, the results are clearly presented and discussed. However, I have some points to be clarify before the publication of the article:
- why did the Authors choose only the 72 h of incubation for cell viability assay? It would be interesting to see also the results for 24 and 48 h. Was it based on any previous preliminary experiments?
- cell viability was determined after 72 h, while the caspase assay after 24 h and apoptosis detection after 24, 48 and 72 h - it would need some comment.
- why most of the experiments was performed on HL60 cells, while the other cell lines were used only for viability assay? This needs some justification.
- the interesting effect of the slight enhancement of silibin and amanitin used together was noted - do the Authors have any idea how to explain it? Especially if the other used hepatoprotectants did not reveal such effect.
Comments on the Quality of English LanguageThe English is good, only some minor errors appear.
Author Response
- why did the Authors choose only the 72 h of incubation for cell viability assay? It would be interesting to see also the results for 24 and 48 h. Was it based on any previous preliminary experiments?
The choice to incubate for 72 hours was indeed based on preliminary experiments. In the experiments no clear effect of the toxin was observed at 24 and 48 hours. Please refer to the figure below for the results from these experiments. Based on these preliminary experiments, we decided to perform the follow up experiments at 72 hours. In addition, we decided to include an additional, higher concentration (10 µM) of α-amanitin as well. We have rewritten the results section to more clearly indicate that no effect of the toxin was observed after 24 and 48 hours (line 84-85 of the revised manuscript). In line, no clear effect of the toxin was observed at cell viability after 24 hours (supplementary figure S3). Only at higher concentrations an effect was observed at 48 hours.
- cell viability was determined after 72 h, while the caspase assay after 24 h and apoptosis detection after 24, 48 and 72 h - it would need some comment.
Cell viability/cell death was used as an end parameter and were measured after 72 h based on preliminary experiments, see answer above. Caspase activity is the initial process prior to apoptosis/cell death. Therefore, any change in caspase activity was expected before the decrease in cell viability and increase in the number of apoptotic cells. Consequently, the caspase assay was conducted after 24 hours. We observed increased caspase activity already after 24 hours, while no effect was observed on the number of cells, shown in Figure S3. As suggested by the reviewer, we have adjusted the corresponding section accordingly (page 5, line 180-183).
- why most of the experiments was performed on HL60 cells, while the other cell lines were used only for viability assay? This needs some justification.
HL60 cells were used for most experiments, because this was the cell line we first started working with. The 5 additional cell lines were used to confirm whether the observed effects were specific for the HL60 cells. For clarification, the corresponding sentence has been rewritten (page 3, line 102-104).
- the interesting effect of the slight enhancement of silibin and amanitin used together was noted - do the Authors have any idea how to explain it? Especially if the other used hepatoprotectants did not reveal such effect.
Thank you for this question. Previous research has shown that silibinin significantly suppressed cell growth and caused apoptosis in HL60 cells (reference – doi: 10.18632/oncotarget.19153). In other studies, N-acetylcystein and benzylpenicillin did not enhance cytotoxicity of other compounds in HL60 cells (references – doi: 10.1016/j.fct.2012.04.014; doi: 10.1007/BF00117857). Based on the literature, we hypothesize that silibinin may have a synergistic effect on α-amanitin-induced toxicity, but the exact underlying mechanism is unknown. The point has been addressed in the discussion section.

Reviewer 2 Report
Comments and Suggestions for Authors
I have read a manuscripy entitled "Unraveling hematotoxicity of α-amanitin, an in vitro study". Before it can be published, it is better for authors to revise the manuscript first by following the comments below:
[General]
1. Please pay attention on the punctuation, typos, and grammatical errors.
2. Please make all taxonomic numenclature in Italic.
3. Figure 1b no 'alpha' on the 'amanitin' at the x-axis
4. All abbreviation should written in its first name, when firstly appeared. For instance, standard deviation (SD).
5. Make sure no errors are derived from the automatic reference manager.
[Title]
1. Too broad. Please be more specific especially on the 'in vitro'
[Abstract]
1. "...cell number and mitochondrial activity in all cell lines...." this sounds off 'cell number.... cell lines' please revise.
2. "α-Amanitin increased caspase-3/7 activity and cleaved caspase-3." Please provide the quantitative data here.
3. "The antidotes and OATP1B3 inhibitors..." what are the antidotes, no explanation in the abstract.
4. PATP1B3 should be given the definition first.
5. Line 16--19 can be made to be more concise because they sound similar.
6. "24, 48, and 72 hours"
[Introduction]
1. "OATP1B3" should be defined first.
2. Please provide the explanation as to why the in vitro approaches are suitable for the study.
3. Novelty of the study should be explicitly stated.
[Methods]
1. Add a section for materials. Please also state how they were collected/procured. Also what are the quality standard of the material? Analytical? Pharmaceutical? Please explain.
2. "the manufacturer’s instructions." state the detail of the assay kit so that readers can reproduce the research.
3."when appropriate." What do authors mean by this? Does that mean being normally distributed or the data type?
4. Authors kept the number of cells below 1,000,000 in each well. But, don't you think that was to much.
5. Why did authors not perform in vivo? What about the other expression of genes/proteins related to hepatoxicity (Reference - doi: 10.2147/CBF.S27901).
[Discussion]
1. Authors claimed that "no effects of these antidotes..." were observed in their study. But authors did not investigate the effect on the molecular target of the antidote. This can be misleading - please revise.
2. "Amatoxins are detected in urine within 90-120 minutes after ingestion" Please add "according to a previous study". Please pay attention to this. Author should actively distinguish the results from their present study and those from previous study to make it clear and less confusing for readers.
3. "Clinically relevant concentrations of α-amanitin...." not sure why authors want to discuss about this. Usually the flow of discussion should be started by the finding of their study, than continued with comparisons, and then the theoretical explanation of the finding. Please consider using this flow. And please remove discussion that is not related (or far-reaching) to the findings.
#Cheers!
Comments on the Quality of English LanguagePlease refer to the general comment above.
Author Response
Response to the comments raised by reviewer 2
[General]
- Please pay attention on the punctuation, typos, and grammatical errors.
The manuscript has been checked thoroughly for these errors.
- Please make all taxonomic numenclature in Italic.
As suggested by the reviewer, we have checked the manuscript and adjusted all taxonomic nomenclature throughout the manuscript.
- Figure 1b no 'alpha' on the 'amanitin' at the x-axis
Because Figure 1B depicts both α-amanitin and β-amanitin, we had decided not to add ‘alpha’ to the x-axis. As suggested by the reviewer, we have replaced ‘amanitin’ by ‘α-/β-amanitin’ in the revised manuscript.
- All abbreviation should written in its first name, when firstly appeared. For instance, standard deviation (SD).
The manuscript has been checked and adjusted accordingly.
- Make sure no errors are derived from the automatic reference manager.
The manuscript has been checked and adjusted accordingly.
[Title]
- Too broad. Please be more specific especially on the 'in vitro'
As suggested by the reviewer, the title has been revised to ‘Unraveling hematotoxicity of α-amanitin in cultured hematopoietic cells’ (page 1, line 2-3).
[Abstract]
- "...cell number and mitochondrial activity in all cell lines...." this sounds off 'cell number.... cell lines' please revise.
This sentence has been rewritten as suggested by the reviewer (page 1, line 13-14).
- "α-Amanitin increased caspase-3/7 activity and cleaved caspase-3." Please provide the quantitative data here.
As suggested by the reviewer, the quantitative data has been included in the abstract (page 1, line 15-17).
- "The antidotes and OATP1B3 inhibitors..." what are the antidotes, no explanation in the abstract.
The antidotes and OATP1B3 inhibitors are defined at the beginning of the abstract (page 1, line 7-10). Due to the word limit of the abstract it is not possible to repeat the antidotes and inhibitors.
- PATP1B3 should be given the definition first.
As suggested by the reviewer, OATP1B3 has been defined at first use (page 1, line 9).
- Line 16--19 can be made to be more concise because they sound similar.
The reviewer is correct, this section has been rewritten to be more concise (page 1, line 18-19).
- "24, 48, and 72 hours"
Unfortunately, it was not fully clear to us what this point was meant to address. We have interpreted it as a suggestion to include the time points of the measurements. We agree with this suggestion and have therefore adjusted the abstract accordingly (page 1, line 7-10).
[Introduction]
- "OATP1B3" should be defined first.
OATP1B3 has been defined where it is first mentioned in the introduction (page 1, line 39-41).
- Please provide the explanation as to why the in vitro approaches are suitable for the study.
Human cells have been used to eliminate any species-related differences. Second, 6 cell lines have been used to represent the range of progenitor cells types and to account for any differences in toxin sensitivity due to differences among progenitor cells (reference – doi: 10.1002/cptx.45). For example, HL60 and K562 cells have been used in previous preclinical research to study hematotoxicity (reference – doi: 10.1097/00001813-200302000-00010). A cell viability assay based on mitochondrial activity has been used to assess dose- and time-dependent toxicity in HL60 cells (reference – doi: 10.3390/ijerph2006030017). Trypan blue assay is widely used to determine cell number and cytotoxic effects of compounds and the MTS assay is commonly used as cell proliferation assay based on mitochondrial activity (reference – doi: 10.2174/1389201017666160808160513). We have shown the relation between mitochondrial activity and cell number in Figure 1. The colony-forming cell assay is commonly used to study hematotoxicity (reference – doi: 10.1002/cptx.45). Caspase-3/7 activity (reference – doi: 10.1007/s12035-023-03433-5), cleaved caspases (reference – doi: 10.1007/978-1-0716-1162-3_1) and annexin V/PI staining (reference – doi: 10.3791/2597) are commonly used in vitro approaches to study apoptosis.
- Novelty of the study should be explicitly stated.
As suggested by the reviewer, we have included this in the introduction (page 2, line 66-67).
[Methods]
- Add a section for materials. Please also state how they were collected/procured. Also what are the quality standard of the material? Analytical? Pharmaceutical? Please explain.
As suggested by the reviewer, a section for materials has been added, including where the materials were procured/purchased and the quality grade of the materials (page 9, line 333-357).
- "the manufacturer’s instructions." state the detail of the assay kit so that readers can reproduce the research.
To address this point, we have included the catalog numbers of the assay kits in the materials section (page 9, line 342 and line 344).
3."when appropriate." What do authors mean by this? Does that mean being normally distributed or the data type?
As suggested by the reviewer, this has been elaborated on which statistical analysis was used for which comparisons (page 10, line 413-415).
- Authors kept the number of cells below 1,000,000 in each well. But, don't you think that was to much.
This high number of cells was required to obtain sufficient lysate for the assay after cell lysis. Each well contained a volume of 2 ml, therefore containing a maximum cell concentration of 500,000 cells/ml. Previous studies describe that HL60 cells reach cell concentrations up to 1,000,000 to 3,000,000 cells/ml in media supplemented with 5-10% foetal calf serum (PMID 3311197). In culture, HL60 cells proliferated with a doubling time of approximately 24-36 hours. Therefore, the maximum cell concentration would not have been reached after 24 hours incubation in the control or samples exposed to α-amanitin. At a similar cell concentration in culture, no increase in percentage of dead cells were observed compared to a lower cell concentration. Therefore, we believe this number of cells is unlikely to have affected the results. We have added the cell concentration for clarification (page 10, line 405).
- Why did authors not perform in vivo? What about the other expression of genes/proteins related to hepatoxicity (Reference - doi: 10.2147/CBF.S27901).
We have initiated these experiments based on our observations in patients, the in vivo results have been submitted in parallel with this manuscript. In the other paper we describe our findings demonstrating that hematotoxicity is also observed in patients. In this manuscript, we did have not performed in vivo experiments, because rodents would not provide a representative model of α-amanitin induced hematotoxicity. There are significant differences in the toxicokinetics of α-amanitin in rodents compared to humans according to literature, e.g. lacking oral bioavailability (reference – doi: 10.1111/j.1399-3011.1983.tb02093.x). Many of the genes and proteins related to hepatotoxicity referenced by the reviewer are only relevant in vivo. However, we agree with the reviewer that these experiments may be done as a next step in future research if this hurdle is overcome.
[Discussion]
- Authors claimed that "no effects of these antidotes..." were observed in their study. But authors did not investigate the effect on the molecular target of the antidote. This can be misleading - please revise.
The reviewer is correct. This sentence has been rewritten accordingly (page 6, line 203-205).
- "Amatoxins are detected in urine within 90-120 minutes after ingestion" Please add "according to a previous study". Please pay attention to this. Author should actively distinguish the results from their present study and those from previous study to make it clear and less confusing for readers.
Thank you for this remark. We have added the phrase as the reviewer suggested (page 7, line 245-246). In addition, the remainder of the discussion has been checked and adjusted accordingly.
- "Clinically relevant concentrations of α-amanitin...." not sure why authors want to discuss about this. Usually the flow of discussion should be started by the finding of their study, than continued with comparisons, and then the theoretical explanation of the finding. Please consider using this flow. And please remove discussion that is not related (or far-reaching) to the findings.
We have added this section to the manuscript to support the clinical relevance of the concentrations of α-amanitin we have used in our experiments, and to put these concentrations into context of existing literature. This section has been adjusted to further clarify this purpose (page 7, line 253-254).
Round 2
Reviewer 2 Report
Comments and Suggestions for Authors
Authors have addressed my concerns appropriately. I recommend the publication of their submitted manuscript.